# Mesenchymal Stem Cell-Derived Extracellular Vesicles Protect Human Corneal Endothelial Cells from Endoplasmic Reticulum Stress-Mediated Apoptosis

**DOI:** 10.3390/ijms22094930

**Published:** 2021-05-06

**Authors:** Lola Buono, Simona Scalabrin, Marco De Iuliis, Adele Tanzi, Cristina Grange, Marta Tapparo, Raffaele Nuzzi, Benedetta Bussolati

**Affiliations:** 1Department of Biotechnology and Health Sciences, University of Turin, 10126 Turin, Italy; lola.buono@unito.it (L.B.); adele.tanzi@edu.unito.it (A.T.); 2Eye Clinic, Department of Surgical Sciences, University of Turin, AOU Città della Salute e della Scienza, Ophtalmic Clinic, 10126 Turin, Italy; scalabrin.simona@gmail.com (S.S.); marco.deiuliis@edu.unito.it (M.D.I.); raffaele.nuzzi@unito.it (R.N.); 3Department of Medical Sciences, University of Turin, 10126 Turin, Italy; cristina.grange@unito.it (C.G.); marta.tapparo@unito.it (M.T.)

**Keywords:** exosomes, corneal endothelium, corneal dystrophy

## Abstract

Corneal endothelial dystrophy is a relevant cause of vision loss and corneal transplantation worldwide. In the present study, we analyzed the effect of mesenchymal stem cell (MSC)-derived extracellular vesicles (MSC-EVs) in an in vitro model of corneal dystrophy, characterized by endoplasmic reticulum stress. The effects of MSC-EVs were compared with those of serum-derived EVs, reported to display a pro-angiogenic activity. MSC-EVs were able to induce a significant down-regulation of the large majority of endoplasmic reticulum stress-related genes in human corneal endothelial cells after exposure to serum deprivation and tunicamycin. In parallel, they upregulated the Akt pathway and limited caspase-3 activation and apoptosis. At variance, the effect of the serum EVs was mainly limited to Akt phosphorylation, with minimal or absent effects on endoplasmic reticulum stress modulation and apoptosis prevention. The effects of MSC-EVs were correlated to the transfer of numerous endoplasmic reticulum (ER)-stress targeting miRNAs to corneal endothelial cells. These data suggest a potential therapeutic effect of MSC-EVs for corneal endothelial endoplasmic reticulum stress, a major player in corneal endothelial dystrophy.

## 1. Introduction

Corneal endothelium, a monolayer of endothelial cells attached to the Descemet membrane [1], is required to maintain corneal transparency and clarity [2]. A loss of corneal endothelial cells leads to vision impairment and corneal edema. In particular, Fuchs’ endothelial corneal dystrophy, a common cause of corneal vision loss, is characterized by endothelial damage as a result of sporadic or autosomal dominant non-inflammatory endothelial dystrophy [3]. So far, there is no definitive conservative non-surgical treatment [4], and this pathology is among the leading causes of primary corneal transplant surgery in Europe and in the United States. Endothelial corneal dystrophy is believed to be caused primarily by endoplasmic reticulum (ER) stress [5]. In particular, the activation of ER stress is known to induce corneal endothelial cell apoptosis, leading to progressive detachment of the cells from the Descemet membrane, starting from the center and spreading to the periphery [5,6].

Intracellular transduction pathways activated by ER stress, collectively termed as unfolded protein response, are considered to be a cellular counteraction to damage aimed at restoring protein folding homeostasis [7]. The ER stress pathways are triggered by three main receptors (PERK, IRE1, and ATF6), which are normally maintained inactivated in the ER through the binding of the chaperone protein GRP78. During ER stress, GRP78 detaches from those receptors, causing their activation and the induction of ER stress related genes [8].

At present, as endothelial corneal dystrophy is only cured by transplantation from a cadaveric donor cornea, it is of relevant importance to find alternative strategies to induce regeneration, survival, and to inhibit ER stress activation in corneal endothelial cells. Extracellular vesicles (EVs) are considered to be a new promising therapeutic tool for tissue regeneration. EVs are present in all biological fluids, including blood [9,10]; they are released by almost all cell types; and they play a central role in cell-to-cell communication through the transfer of their cargo to target cells [11,12,13]. Among different EV sources, mesenchymal stem cell (MSC) derived EVs possess unique anti-inflammatory, anti-apoptotic, tissue repairing, proangiogenic, and immunomodulatory properties, similar to their parental cells [14,15]. In the last decade, MSC-EVs have been of interest in many ophthalmologic pathologies, from retinal damage to corneal diseases, with promising emerging findings [14,16,17,18,19]. For instance, corneal MSC-EVs ameliorated corneal epithelial damage, showing an increased proliferation of human corneal epithelial cells in vitro, and accelerated corneal wound closure in a murine epithelial mechanical injury model [17]. Our group recently reported the proangiogenic effect of MSC-EVs in an in vitro model of corneal endothelium damage due to serum deprivation. MSC-EVs induced the proliferation and survival of damaged human corneal endothelial cells (HCECs), and promoted their wound closure [20].

In the present study, we first aimed to study MSC-EVs’ effects on HCECs, investigating their effect in in vitro models of ER stress induced apoptosis, as well as the possible mechanisms involved. In particular, we used both serum starvation and tunicamycin, a specific ER-stressor, to examine the effects of MSC-EVs on ER stress signaling pathways. Moreover, we compared the effects of MSC-EVs with that of blood serum-derived EVs (SER-EVs), isolated from healthy donors, already reported to induce angiogenesis in vitro and repair from acute ischemia in vivo [21]. Finally, we examined the MSC-EV microRNA cargo involved in the modulation of the ER stress pathways, and its transfer to the target cells.

## 2. Results

### 2.1. Isolation and Characterization of Human Corneal Endothelial Cells and EVs

HCECs were isolated and characterized as previously described [20]. In brief, we isolated HCECs from discarded corneas of patients undergoing corneal transplantation or enucleation (N = 23 patients; Appendix A). A cell line of HCECs was isolated from each patient; all of the experiments were performed between passage 2 and 4. The HCECs were characterized by morphology, showing cells with an organized cytoskeletal structure and with the presence of the typical corneal endothelial markers [1]: CD166, Na^+^K^+^ATPase, and ZO-1 (Figure 1A–C) [22].

MSC-EVs were isolated from the supernatants of MSCs cultured overnight in FCS-deprived RPMI, and SER-EVs were isolated from 100 mL of serum obtained from a blood pool of five healthy donors. The MSC supernatant and serum were then centrifuged for the removal of cell debris and apoptotic bodies at 3000g for 20 min. EVs were purified by 2 h of ultracentrifugation at 100,000g at 4 °C [21,23]. EVs were analyzed by size distribution and were quantified using the nanoparticle tracking analysis. MSC-EVs and SER-EVs appeared as homogenous EV populations with a mean size of 159.6 nm and 238.9 nm, respectively (Figure 2A). The EV size, morphology, and tetraspanin expression were assessed using super resolution fluorescence microscopy (Figure 2B), confirming the simultaneous presence of tetraspanins on the surface of EVs derived from both sources. CD9 and CD81 appeared in a clustered, uneven distribution in the EV membrane, at a variance of CD63. This could be related to the reported ability of tetraspanins to homodimerize and form large complexes [24]. In addition, EVs were characterized by their surface marker expressions, including tetraspanins, mesenchymal, endothelial/platelet, and immunological markers, using a MACSPlex Exosome analysis kit after bead-based immunocapture (Figure 2C–E). The levels of tetraspanins CD9, CD63, and CD81 were high in both of the EV types (Figure 2E). The EVs isolated from the serum expressed immune system markers such as CD40, CD8, CD24, and HLA-DR (Figure 2E), while MSC-EVs showed higher levels of class-I HLA-ABC. MSC typical markers CD105, CD146, CD44, SSEA4, CD29, and CD49e were confirmed in the MSC-EVs (Figure 2E). Endothelial and platelet markers CD62P, CD42a, CD69, CD31, and CD41b were in turn typically expressed in the EVs derived from the serum (Figure 2E).

### 2.2. ER Stress Induction by Serum Deprivation and Tunicamycin in HCECs

In order to set the model of ER stress in HCECs, we first exposed HCECs to different doses of tunicamycin. In fact, this drug is known to inhibit the first step in the N-linked glycan biosynthesis of proteins, thus inducing misfolded proteins and an unfolded protein response [25]. When tunicamycin was applied to HCECs, the expression levels of the main ER stress-related proteins GRP78 [26] and C/EBP homologous protein (CHOP) [27] were significantly increased in a dose-dependent manner (Figure 3). In parallel, we used the serum deprivation model of HCEC damage already set for previous experiments [20]. In this model, ER stress related genes were upregulated in a comparable manner to the tunicamycin treatment, confirming nutrient deprivation as an inducer of ER stress. The GAPDH levels were similar between the samples in each experiment.

### 2.3. ER Stress Induction by Serum Deprivation and Tunicamycin in HCECs

We subsequently analyzed the effect of MSC-EVs and SER-EVs on the ER stress damage model described above, by evaluating the typical ER stress-related genes ATF4, GRP78, XBP1t, and CHOP [5,25,26]. Cells were deprived of serum or treated with tunicamycin for 24 h, and subsequently stimulated with MSC-EVs or SER-EVs for a further 3 h (Figure 3C). The lower concentration of tunicamycin (0.4 ng/mL) was used, as the results were comparable to those obtained with serum deprivation. As expected, a significant upregulation of these ER stress-related genes was observed when the HCECs were deprived of serum or treated with tunicamycin in comparison with the untreated HCECs (CTL; Figure 4). Moreover, MSC-EVs were able to induce a significant down-regulation of the large majority of ER stress-related genes. In particular, ATF4, GRP78, XBP1t, and CHOP were all reduced in HCECs after exposure to serum deprivation and tunicamycin (Figure 4). At variance, SER-EVs only partially recovered the ER stress damage, modulating ATF4 in serum deprivation and GRP78 in tunicamycin exposure (Figure 4). The GAPDH levels were similar between the samples in each experiment.

We then evaluated the regulation of EIF2a, an activator of the ER stress pathway [28], in the model of serum deprivation. We selected serum deprivation over tunicamycin treatment because of the more consistent results obtained. EIF2a phosphorylation was significantly induced by serum deprivation, and inhibited by the presence of both MSC-EVs and SER-EVs (Figure 5A).

Finally, we considered that the ER stress pathway is linked to Akt regulation, with a reciprocal inhibition, resulting in reduced cell proliferation and induced apoptosis [29]. Given this, we analyzed the levels of Akt phosphorylation during serum deprivation and subsequent treatment with MSC-EVs and SER-EVs (Figure 5B). The significant reduction of Akt phosphorylation after HCEC exposure to serum deprivation was significantly restored by both MSC-EVs and SER-EVs treatment (Figure 5B). Altogether, these data may suggest a differential effect of MSC-EVs and SER-EVs on ER stress gene and Akt pathway modulation.

### 2.4. MSC-EVs, but Not SER-EVs, Reduce HCECs Apoptosis

We previously showed that serum deprivation could induce HCEC apoptosis and inhibit cell proliferation, and demonstrated that MSC-EVs were able to restore this type of damage [17]. Here, we compared the effect of SER-EVs to that of MSC-EVs. Serum deprivation, as expected, resulted in a significant, although not massive, increase in the total apoptotic cells (Figure 6A and Appendix A) and a decrease in live cells (Figure 6B and Appendix A). Treatment with MSC-EVs re-established the basal levels of the control cells (Figure 6A and Appendix A) and increased cell viability (Figure 6B and Appendix A). Similar results were obtained with tunicamycin exposure. Accordingly, only the treatment with MSC-EVs was able to reduce the levels of the pro-apoptotic caspase-3 in stress conditions (Figure 6C,D).

### 2.5. MSC-EV miRNAs Transfer to HCECs

We previously analyzed the miRNA content of both MSC-EVs and SER-EVs [30,31]. In order to explore the mechanisms that account for the MSC-EV biological activity, we performed a Funrich analysis, combining the mostly expressed miRNAs together with the miRNAs predicted to target the main ER stress genes ATF4, CHOP, and XBP1. We found that several miRNAs targeting the studied genes were also part of the MSC-EVs cargo (Figure 7A). Among those, we focused on the miRNAs with the highest expression in MSC-EVs [30,32] and targeting all three ER-stress genes (Figure 7B), and on miR-214-3p, which was not predicted by miRwalk but was observed by extended literature to target key genes of ER stress [33,34]. The expression levels of the most relevant miRNAs were found to be lower in SER-EVs, with the exception of miR199a-3p (Figure 7B). We subsequently evaluated the possible transfer of ER stress targeting miRNAs to serum deprived HCECs treated with MSC-EVs and SER-EVs, in the presence of amanitin, in order to block endogenous transcription. Figure 7 shows that the miRNA levels found in HCECs were significantly higher in cells exposed to MSC-EVs (Figure 7C). miR-199a-3p was significantly upregulated in both EV treatments, in accordance with its comparable Ct value in both EV sources (Figure 7B).

## 3. Discussion

In the present study, we analyzed the effect of MSC-EVs and SER-EVs on the activation of ER stress genes in endothelial cells derived from human corneas. We found that MSC-EVs had a profound effect on the ER stress pathway at different levels, including GRP78, ATF4, CHOP, and XBP1 induction, as well as EIF2a phosphorylation, and prevented cell apoptosis. SER-EVs, at variance, showed a minor effect on the ER stress pathway, and only a specific effect on Akt activation, without preventing cell apoptosis.

ER stress is a relevant pro-apoptotic player in corneal endothelial cells [5] and, in particular, has a pathogenic role in genetic corneal endothelial dystrophy, one of the most frequent needs for corneal transplantation worldwide [35]. The cornea is an ideal organ for regenerative cell therapy, because of its immune-privileged and avascular nature [36]. Therefore, MSCs and their bioproducts, such as EVs, have received much attention among ophthalmologists and visual scientists as an alternative way to manage corneal diseases [22,37,38]. The in vivo treatment with MSC-EVs has been successfully studied in several diseases of the retina, such as retinal cell degeneration, refractory macular holes, and retinal detachments [18,39]. MSC-EV administration typically takes place through intravitreal injection, allowing their direct action on the target cells.

In the present study, in an in vitro model of ER stress, we found that MSC-EVs were highly active, at both of the doses studied, on all the most typical genes induced by ER-stress, namely: ATF4, GRP78, XBP1, and CHOP [5,6,25,40]. Moreover, we found that MSC-EVs significantly reduced the apoptosis of corneal endothelial cells in parallel with EIF2a inhibition and caspase-3 downregulation. These data suggest that the anti-apoptotic effect of MSC-EVs in our in vitro models could be directly related to the lowering of ER stress. Our results are in line with the well-established effect of MSC-EVs on the amelioration of ER stress activation shown in a number of different cell types, including tubular epithelial cells, neural cells, and pancreatic beta cells [41,42,43]. Moreover, as described in other models [44,45,46], we found that MSC-EVs could act in combination with the Akt pathway activation in corneal endothelium. The restoration of the levels of Akt phosphorylation resulted in a significant downregulation of the pro-apoptotic caspase-3, in line with the reduced number of apoptotic HCECs following treatment with MSC-EVs.

Furthermore, a recent study showed that exosomes from blood plasma could activate Akt to regulate angiogenesis and promote the expression of anti-apoptotic proteins [47]. In accordance with these recent findings, we analyzed the activity of EVs derived from serum, a very accessible source. We were able to confirm that EVs derived from the blood serum of healthy donors could restore the levels of Akt impaired by ER stress, resulting in the regulation of one ER stress related gene, ATF4. However, the modulation of ER stress genes was far lower than that of MSC-EVs. Moreover, SER-EVs, as well as MSC-EVs, were able to significantly reduce the levels of EIF2, a key protein phosphorylation in ER stress known to be a counter-regulator of Akt. However, SER-EVs were not able to restore the levels of caspase-3, nor to reduce the apoptotic number of HCECs. These data highlight a possible therapeutic effect of MSC-EVs, and not of SER-EVs, on endothelial corneal ER stress activation.

The functional effects of MSC-EVs on tissue repair are mainly ascribed to the transfer of their miRNA cargo [32]. Here, we found that a large number of miRNAs, previously involved in the regulation of the different ER stress related genes [33,41,43], were present in MSC-EVs, suggesting a strict connection of those miRNAs with the observed ER stress regulation. Indeed, the highest expressed MSC-EV miRNAs involved in ER stress were found to be transferred in HCECs. The absence of transfer by SER-EVs confirms the specificity of MSC-EV miRNA transfer, and of the related effect on the apoptosis of corneal endothelial cells. Other studies have shown that miRNAs may act to reduce the phosphorylation of EIF2a, decreasing the translation of ATF4, which inhibits the expression of CHOP, thereby resulting in cell survival and apoptosis reduction [48]. Our results showing the upregulation of several miRNAs due to MSC-EV treatment suggest that the observed miRNA transfer is responsible for the MSC-EV mediated reduction of the levels of ER stress and apoptosis. However, the mechanisms underlying the miRNA-dependent regulation of ER stress genes have not been fully elucidated yet.

In conclusion, our study reveals a novel effect of MSC-EV on corneal ER stress-induced cell death, with possible implications in corneal endothelial dystrophy. These data may pave the road for designing more effective therapeutic strategies to battle corneal endothelial pathologies, possibly through direct intra-aqueous humor injection.

## 4. Materials and Methods

### 4.1. Isolation and Characterization of HCECs

Human corneal endothelial cells (HCECs) were isolated and characterized as previously described [20]. The study was approved by the Intercompany Ethics Committee of A.O.U. Città della Salute e della Scienza (Turin), on 21 February 2020 (reference number 00184/2019). Informed written consent was obtained from all of the tissue donors. In brief, we isolated HCECs from the discarded corneas of patients undergoing corneal transplantation or enucleation (N = 23 patients). The Descemet’s membrane and corneal endothelial cells were stripped from the posterior surface of the peripheral corneoscleral tissue using a scalpel, and afterwards, was digested with collagenase A (2 mg/mL). The digested membrane and cells were then placed on a Petri dish previously coated with fibronectin. HCECs migrated out of the Descemet’s membrane and were maintained at 37 °C in a humidified atmosphere of 95% air and 5% CO_2_, and were cultured in ENDOGRO (MilliporeSigma™, Burlington, MA, USA) supplemented with 10% FBS (not deprived of EVs). Once confluency was reached, they were passaged at a 1:2 ratio using a 0.25% trypsin and 0.02% ethylenediaminetetraacetic acid solution. From each patient, we isolated a cell line of HCECs, each cell line was kept in culture, and all of the experiments were performed between passage 2 and 4. The cells were characterized by immunofluorescence using phalloidin antibody (ThermoFisher Scientific, Waltham, MA, USA) and DAPI for nuclear staining, for the expression of HCEC main marker Na+K+ATPase, ZO-1, and by presence of the surface marker CD166, as described [20].

### 4.2. Isolation and Characterization of MSC-EVs and SER-EVs

Human bone marrow-derived MSCs were obtained from Lonza (Switzerland), characterized as described [22], and cultured until the sixth passage. All of the cell preparations used were positive for the typical MSC markers (CD105, CD29, CD73, CD44, and CD90). MSC-EVs and SER-EVs were obtained and characterized as previously described [23,31]. Briefly, MSC-EVs were obtained from the supernatants of MSCs cultured overnight in RPMI deprived of FCS. SER-EVs were obtained from a total of 100 mL of serum isolated from a blood pool of five healthy donors. Informed consent was obtained by the Blood Bank of “Città della Salute e della Scienza di Torino” from all of the participants. MSC supernatant and serum were then further centrifuged for the removal of cell debris and apoptotic bodies at 3000g for 20 min. EVs were purified for 2 h ultracentrifugation at 100,000g at 4 °C (Beckman Coulter, Brea, CA, USA). EVs from both sources were used freshly or stored at −80 °C after resuspension in RPMI supplemented with 1% dimethyl sulfoxide. MSC and serum-derived EVs were characterized by MACSPlex flow cytometry for the expression of CD63, CD81, and CD9, and the main mesenchymal, platelet, and endothelial surface markers. Analysis of the size distribution and enumeration of EVs were performed using NanoSight NS300 (NanoSight Ltd, Malvern, UK) equipped with a 405 nm laser and the Nanoparticle Tracking Analysis (NTA) 2.3 software (NanoSight Ltd., Malvern, UK).

### 4.3. MACSPlex Analysis

Different sample types were subjected to bead-based multiplex EV analysis by flow cytometry (MACSPlex Exosome Kit, human, Miltenyi Biotec, Auburn, CA, USA), and the EV-containing samples were processed as previously described [49]. Samples were diluted with a MACSPlex buffer (MPB) to a final volume of 120 µL. Then, 15 µL of MACSPlex Exosome Capture Beads (containing 39 different antibody-coated bead subsets) were added to each sample. Generally, particle counts quantified by NTA, and not protein amount, were used to estimate the input EV amounts. The samples were then incubated on an orbital shaker overnight (14–16 h) at 450 rpm at −4 °C, and were protected from light. To wash the beads, 1 ml of MPB was added and removed after several centrifugations (3000g, 5 min). For counterstaining the EVs bound by capture beads with detection antibodies, 135 µL of MPB and 5 µL of each APC-conjugated anti-CD9, anti-CD63, and anti-CD81 detection antibodies were added to each sample and were incubated on an orbital shaker at 450 rpm protected from light for 1 h at room temperature. Next, to wash the beads, 1 mL of MPB was added and removed after one centrifugation (3000g, 5 min). This was followed by another washing step with 200 µL of MPB, incubation on an orbital shaker at 450 rpm protected from light for 15 min at room temperature, and then the MPB was removed. Subsequently, 150 µL of MPB was added to each sample and flow cytometric analysis was performed using FACS Celesta (BD Biosciences, Franklin Lakes, NJ, USA).

### 4.4. Super-Resolution Microscopy

Super-resolution fluorescent microscopy analyses were performed using a Nanoimager S Mark II microscope from ONI (Oxford Nanoimaging, Oxford, UK) equipped with 405 nm/150 mW, 473 nm/1 W, 560 nm/1 W, 640 nm/1 W lasers, and dual emission channels split at 640 nm, as previously described [50]. For the preparation of the sample, 10 μL of Poly-l-Lysine (Sigma-Aldrich, St. Louis, MO, USA) was placed on coverslips, in culture wells (Grace Bio-Labs, Sigma-Aldrich, St. Louis, MO, USA), and left at 37 °C in a humidifying chamber for 2 h. After removal of the excess, MSC-EVs and SER-EVs were left to attach overnight at 4 °C on the coverslips. The next day, non-attached EVs were removed and 10 μL of blocking buffer (PBS-5% Bovine Serum Albumin) was added into the wells for 30 min. Then, mouse anti-CD63, anti-CD81, and anti-CD-9 antibodies (Santa Cruz Biotechnology, Dallas, TX, USA) were conjugated with Alexa Fluor 555, 647, and 488 dyes, respectively, using the Apex Antibody Labeling Kit (Invitrogen, Carlsbad, CA, USA), according to the manufacturer’s protocol. The antibodies were left for overnight incubation at 4 °C. The samples were washed twice with PBS and a 10 μL ONI BCubed Imaging Buffer was added for the EV imaging. Two-channel dSTORM data were acquired sequentially at 30 Hz (Hertz) in the total reflection fluorescence (TIRF) mode. Single molecule data were filtered using NimOS (Version 1.7.1.10213, ONI, Oxford, UK), based on the point spread function shape, photon count, and localization precision to minimize background noise and remove low-precision localizations.

### 4.5. Cytofluorimetric Analysis

For the cytofluorimetric analysis, the cells were detached using a non-enzymatic cell dissociation solution (Sigma-Aldrich, St. Louis, MO, USA), resuspended in PBS 0.1% BSA (Sigma-Aldrich, St. Louis, MO, USA), and incubated with antibodies. For each staining, 100,000 cells were incubated for 20 min at 4 °C with PE-conjugated CD166 antibody (BD Biosciences, Franklin Lakes, NJ, USA). PE-conjugated mouse IgG (Miltenyi Biotec, Bergisch Gladbach, Germany) was used as a negative control. FACS analysis was performed by FACS Celesta cytofluorimeter (BD Biosciences, Franklin Lakes, NJ, USA).

### 4.6. Apoptosis Assay

Apoptosis was evaluated by Muse™ Annexin V and Dead Cell Assay (Millipore, Merck KGaA, Darmstadt, Germany) according to the manufacturer’s instructions. The assay was based on the detection of phosphatidylserine on the surface of apoptotic cells, using fluorescently labeled Annexin V in combination with the dead cell marker, 7-AAD. Briefly, 30 × 10^3^ cells in a 24-well were deprived of serum (2% FBS) or were incubated with 0.4 ng/mL of tunicamycin for 24 h. MSC-EVs or SER-EVs (20 × 10^3^ EV/cell) were added to the medium for a further 24 h. The cells were then detached and resuspended in Muse^TM^ Annexin V and a Dead Cell kit, and the percentages of the total live cells (negative for Annexin V and dead cell marker) and of the apoptotic cells (Annexin V^(+)^ and dead cell marker^(−)^) were detected.

### 4.7. RNA Isolation and Real Time PCR

To perform the mRNA evaluation of the ERstress markers, HCECs under serum deprivation (2% FBS) or tunicamycin (0.4 ng/mL, ENZO Life Sciences, Farmingdale, NY, USA) for 24 h were treated with MSC-EV or SER-EV (10–20 × 10^3^/ cell). After 3 h of EV exposure, the total RNA was isolated using Trizol Reagent (Ambion, Austin, TX, USA) according to the manufacturer’s protocol. RNA was then quantified spectrophotometrically (Nanodrop ND-1000, Wilmington, NC, USA). For the gene expression analysis, quantitative real-time PCR was performed. Briefly, one-strand cDNA was produced from 200 ng of total RNA using a High Capacity cDNA Reverse Transcription Kit (Applied Biosystems, Waltham, MA, USA). Real-time PCR experiments were performed in a 20-μL reaction mixture containing 5 ng of cDNA template, the sequence-specific oligonucleotide primers (purchased from MWG-Biotech, High Point, NC, USA), and the Power SYBR Green PCR Master Mix (Applied Biosystems, Waltham, MA, USA). GAPDH mRNA were used to normalize the RNA inputs. The fold change expression with respect to the control was calculated for all of the samples. The specific sequences of the primers used are listed in the supplementary material (Appendix A).

### 4.8. Protein and Western Blot

For the protein analysis, HCECs were lysed at 4 °C for 30 min in a RIPA buffer (20 nM Tris·HCl, 150 nM NaCl, 1% deoxycholate, 0.1% SDS, 1% Triton X-100, pH 7.8), supplemented with a protease and phosphatase inhibitor cocktail and PMSF (Sigma-Aldrich, St. Louis, MO, USA). Aliquots of the cell lysates containing 25 μg protein, as determined by the Bradford method, were run on 4–20% gel (Biorad, Hercules, CA, USA) under reducing conditions and blotted onto PVDF membrane filters using the iBLOT system (Life Technologies, Waltham, MA, USA). The following antibodies were used: Na^+^K^+^ATPase (#ab2871, Abcam, Cambridge, UK); EIF2a (#9722s), pEIF2a (#9721), Akt (#9272s) and pAkt (#4058s) antibodies, all purchased from Cell signaling, Danvers, CA, USA, ZO-1 (#sc-33725), caspase-3 (#sc-7148), actin (#sc-1616), and vinculin (#sc-7648), all purchased from Santa-Cruz Biotechnology, Dallas, TX, USA.

### 4.9. miRNA Target Prediction

The predicted miRNA targets were obtained from miRWalk (http://mirwalk.umm.uni-heidelberg.de/, accesed on 1 March 2021). As datasets, the list of miRNAs predicted to target ATF4, CHOP, and XBP1 was used. Published data of our laboratory were utilized for comparison with the MSC-EVs miRNA [30] and SER-EVs miRNA content [31]. Data were further analyzed using Expression Suite and Funrich V3 Software (Bundora, Australia).

### 4.10. miRNA Isolation and Analysis

For the miRNA analysis, miScript SYBR Green PCR Kit (QIAGEN, Hilden, Germany) was used. Cells under 2% FBS starvation were treated for 3 h with MSC-EVs or SER-EVs in the presence of the transcriptional inhibitor α-amanitin (50 μg/mL). The RNA samples of the HCECs were reverse transcribed using the miScript Reverse Transcription Kit (QIAGEN, Hilden, Germany), and cDNA samples were used to quantify the miRNAs of interest. Experiments were run using 10 ng of cDNA for each reaction, as described by the manufacturer’s protocol (QIAGEN, Hilden, Germany). RNU6B was used as the endogenous control. Untreated cells (CTL), set as 1, were used as the reference sample for each experiment. The analysis shows the RQ average (2^−^^ΔΔCt^) of at least three independent experiments ± SD. One-way ANOVA with Tukey’s multiple comparisons test was performed. The sequence-specific oligonucleotide primers used are listed in the supplementary material (Appendix A).

### 4.11. Statistical Analysis

One-way ANOVA analysis with Tukey’s multiple comparisons test was performed. A *p* value of < 0.05 was considered to be significant (GraphPad, San Diego, CA, USA).

## Figures and Tables

**Figure 1 ijms-22-04930-f001:**
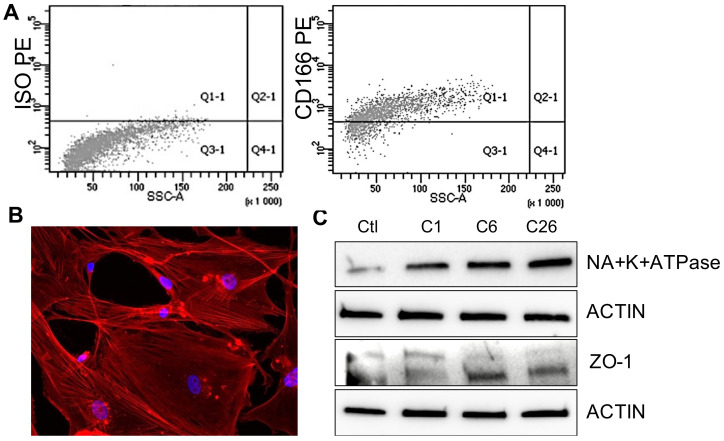
Isolation and characterization of human corneal endothelial cells. (**A**) Representative flow cytometry analysis of human corneal endothelial cells (HCECs) showing the negative staining of a control isotype (ISO PE) and the positive expression of CD166. (**B**) Representative immunofluorescence micrographs of HCECs stained with phalloidin (red), showing an elongated morphology. Blue = nuclear stain DAPI, original magnification: ×20. (**C**) Representative Western blot images of three HCEC independent cell lines positive for Na^+^K^+^ATPase and ZO-1. The renal HK2 cell line was used as a reference (Ctl). Actin was used as an endogenous loading reference.

**Figure 2 ijms-22-04930-f002:**
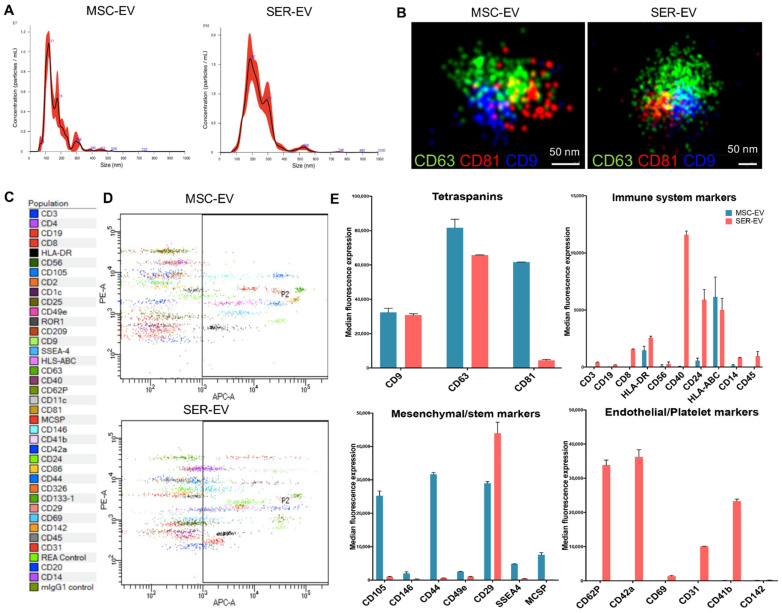
Characterization of mesenchymal stem cell-derived extracellular vesicles (MSC-EVs) and blood serum-derived EVs (SER-EVs). (**A**) Representative nanoparticle tracking analysis showing the EV size distribution. (**B**) Super resolution microscopy micrographs showing the pattern distribution of CD63 in green, CD81 in red, and CD9 in blue for one MSC-EV and SER-EV. Scale bar: 50 nm. (**C**) Legend showing the 39 antibodies used in the assay and their respective colors in the dot plots. (**D**) MACSPlex representative dot plots showing the MSC-EV and SER-EV distribution of allophycocyanin (APC)-stained bead populations; captured EVs are counterstained with APC-labeled detection antibodies using a mixture of anti-CD9, anti-CD63, and anti-CD81 (pan tetraspanins) antibodies. (**E**) Representative quantification of the median APC fluorescence positive values for the bead populations after background correction, clustered in different graphs according to their classification: tetraspanins, immunological, mesenchymal, and endothelial markers.

**Figure 3 ijms-22-04930-f003:**
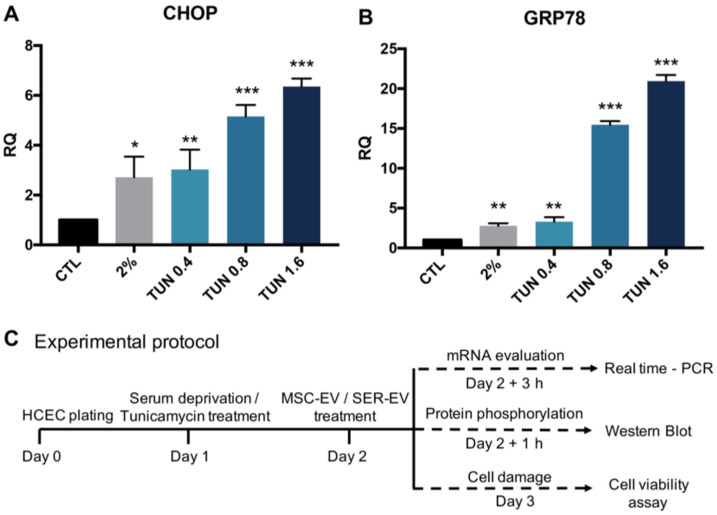
(**A**) Tunicamycin and (**B**) serum deprivation both induce ER-stress. HCECs were treated with 2% Fetal bovine serum (FBS) (2%) and with increasing doses of tunicamycin (TUN 0.4–1.6 ng/mL) for 24 h. The mRNA levels of C/EBP homologous protein (CHOP) and GRP78 significantly increased in all of the damage conditions, when compared with the untreated cells (CTL). Glyceraldehyde 3-phosphate dehydrogenase (GAPDH) was used as an endogenous normalizer. Data were further normalized to CTL, set as 1, and used as a reference sample for each experiment. The graphs show the RQ average (2^−^^ΔΔCt^) of at least three independent experiments ± standard deviation (SD). One-way analysis of variance (ANOVA) with Tukey’s multiple comparisons test was performed after the normalization of each experiment to CTL; * *p* < 0.05, ** *p* < 0.01, and *** *p* < 0.0001 with respect to CTL. (**C**) Experimental protocol used in the different experiments involving MSC- and SEV-EV treatments.

**Figure 4 ijms-22-04930-f004:**
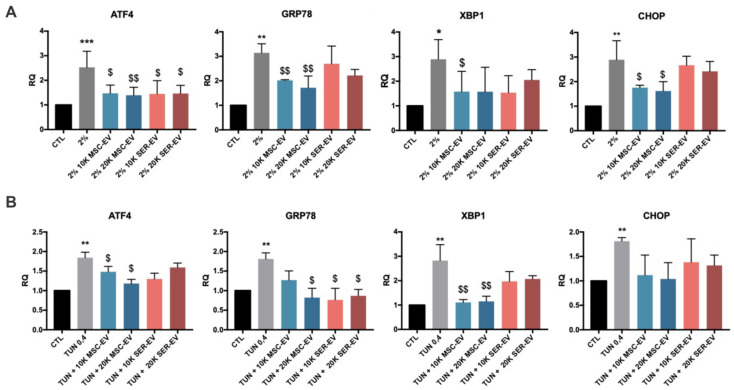
Regulation of the expression of ER stress related genes. HCECs were exposed to serum starvation or tunicamycin for 24 h, and were further stimulated with MSC- or SER-EVs for 3 h. Real time analysis showing the expression of ATF4, GRP78, XBP1, and CHOP with HCECs in normal culture conditions (CTL), in serum deprivation (2%) (**A**), or treated with 0.4 ng/mL of tunicamycin (TUN 0.4) (**B**), with or without 10–20 × 10^3^ MSC-EV/cell (10K/20K MSC-EV) or 10–20 × 10^3^ SER-EV/cell (10K/20K SER-EV). GAPDH was used as an endogenous normalizer. Data were further normalized to CTL, set as 1, and used as a reference sample for each experiment. The graphs show the RQ average (2^−^^ΔΔCt^) of at least three independent experiments ± SD. One-way ANOVA with Tukey’s multiple comparisons test was performed after the normalization of each experiment to CTL; * *p* < 0.05, ** *p* < 0.01, and *** *p* < 0.0001 with respect to CTL, $ *p* < 0.05, $$ *p* < 0.01 vs. 2% (**A**) or TUN (**B**).

**Figure 5 ijms-22-04930-f005:**
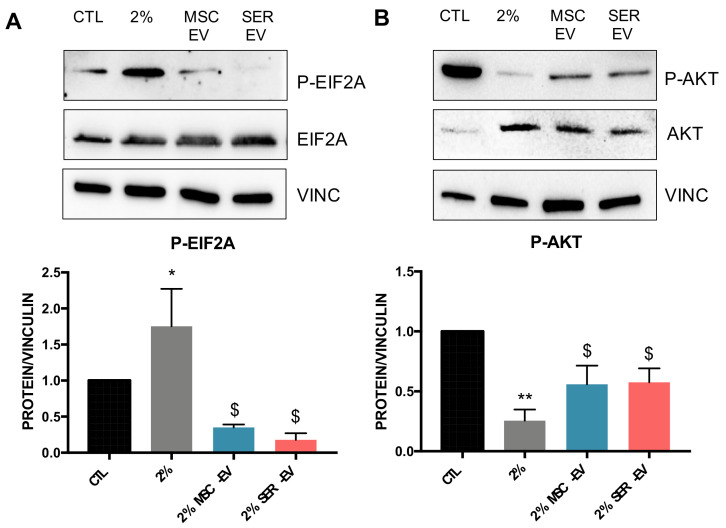
Western blot analysis on HCECs of ER stress related protein phosphorylation. HCECs were exposed to serum starvation for 24 h and further stimulated for 1 h with MSC- or SER-EVs. (**A**,**B**) Representative images of Western blot on HCECs in normal conditions (CTL), in serum deprivation (2%), and in serum deprivation in the presence of 10 × 10^3^ MSC-EV/cell (2% MSC-EV) or in the presence of 10 × 10^3^ SER-EV/cell (2% SER-EV). (**A**) Protein levels and the quantification of phospho-EIF2a normalized to total EIF2a and vinculin (VINC). (**B**) The protein levels and the quantification of p-AKT normalized to total Akt and vinculin (VINC) and to CTL. CTL, set as 1, was used as a reference sample for each experiment. The graphs show the average of at least three independent experiments ± SD. One-way ANOVA with Tukey’s multiple comparisons test was performed after the normalization of each experiment to CTL; * *p* < 0.05 vs. CTL, ** *p* <0.001 vs. CTL, $ *p* < 0.05 vs. 2%.

**Figure 6 ijms-22-04930-f006:**
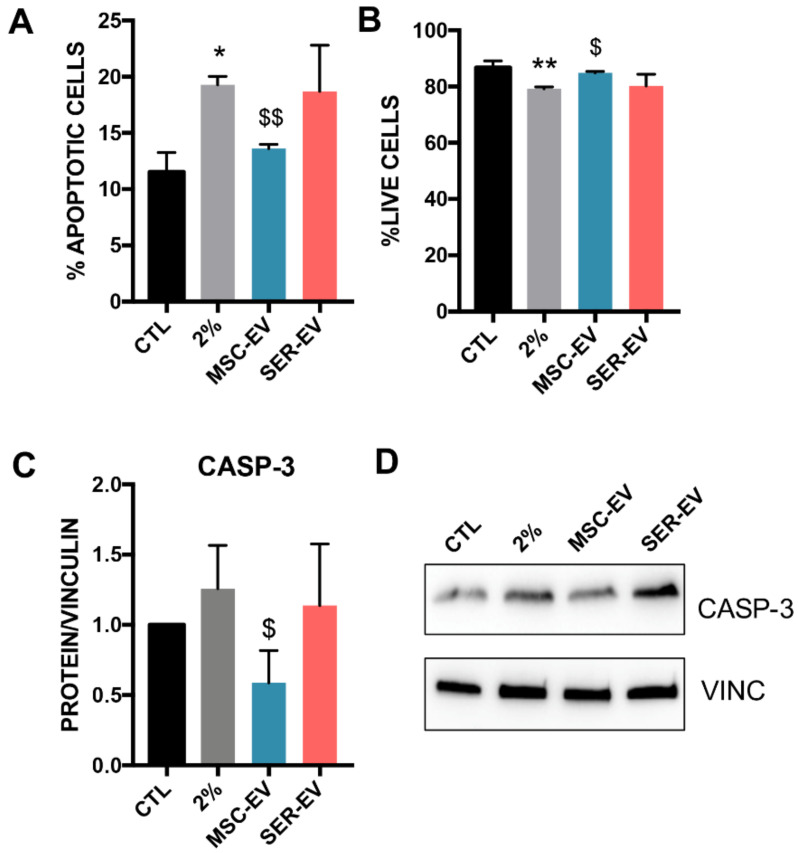
Apoptosis regulation HCECs after serum deprivation and EV treatment. HCECs were exposed to serum starvation for 24 h and further stimulated with MSC- or SER-EVs for 24 h. (**A**) Percentage of total apoptotic HCECs cultured either in normal conditions (CTL) or in serum deprivation for 24 h (2%), and treated for further 24 h with MSC-EVs or SER-EVs (20 × 10^3^ EV/cell). (**B**) Percentage of total live HCECs cultured either in normal conditions (CTL) or in serum deprivation for 24 h (2%), and treated for a further 24 h with MSC-EVs or SER-EVs (20 × 10^3^ EV/cell). (**C**) Protein levels of caspase-3 (CASP-3) and its quantification normalized to total vinculin (VINC). Vinculin was used as the endogenous control. CTL, set as 1, was used as the reference sample for each experiment. The graphs show the average of at least three independent experiments ± SD. One-way ANOVA with Tukey’s multiple comparisons test was performed after the normalization of each experiment to CTL. * *p* < 0.05 vs. CTL, ** *p* <0.001 vs. CTL, $ *p* < 0.05 vs. 2%, $$ *p* < 0.001 vs. 2%. (**D**) Representative image of Western blots on HCECs blotted with caspase-3 and the endogenous control vinculin.

**Figure 7 ijms-22-04930-f007:**
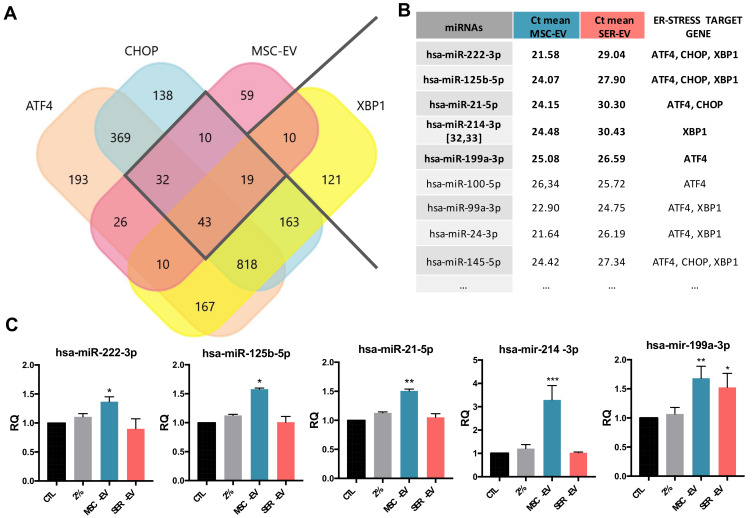
Target miRNAs prediction of MSC-EVs and ER stress related genes. (**A**) Representative Venn diagram showing the numbers of the target miRNAs predicted for CHOP, ATF4, and XBP1, and those described in the literature for MSC-EVs [30]. The numbers displayed in the diagram represent the number of miRNAs that were found to target the indicated transcripts. (**B**) Table showing the miRNAs and Ct means of the most expressed miRNAs in MSC-EVs [29], compared with the Ct values of the same miRNAs in the SER-EVs [31]. Data are displayed as the mean CT value. The ER stress target gene column highlights the target genes of each miRNA among the genes used (ATF4, CHOP, and XBP1). (**C**) Real time PCR analysis of miRNAs transferred by MSC-EVs in HCECs. HCECs were treated with EVs in the presence of amanitin (50 μg/mL). CTL—HCECs in normal medium; 2%—HCECs in serum deprivation; MSC-EVs and SER-EVs—HCECs treated with MSC-EVs and SER-EVs (20 × 10^3^ EV/cell), respectively. RNU6B was used as the endogenous normalizer. Data were further normalized to CTL, set as 1, and used as reference sample for each experiment. The graphs show the RQ average (2^−^^ΔΔCt^) of at least three independent experiments ± SD. One-way ANOVA with Tukey’s multiple comparisons test was performed after the normalization of each experiment to CTL; * *p* < 0.05 vs. CTL, ** *p* < 0.001 vs. CTL, *** *p* < 0.0001 vs. CTL.

## Data Availability

The data presented in this study are available on request from the corresponding author.

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
