# Peer review of "Mesenchymal Stem Cell-Derived Extracellular Vesicles Protect Human Corneal Endothelial Cells from Endoplasmic Reticulum Stress-Mediated Apoptosis"

_ijms, 2021, doi:10.3390/ijms22094930_

Round 1

Reviewer 1 Report

Dear Authors,

I find your paper to be interesting and scientifically sound. Experiments seem well designed and conducted. I just have a "big" question regarding the statistics behind some experiments, plus some elucidations about figure 7. Please, take all the stuff on statistics more as a “clarification needed” than as a critic to the paper, which, again, I find to be very interesting. Please, find below my comments and suggestions for your work.

Minor concerns

1) For unknown reasons, the manuscript came to me with some sections highlighted in yellow. I don't know if it is an uploading/copyediting issue or it is intentional. 
2) Lines 92-107: English language of this section could be improved.
3) Lines 93-94: "nanosight analysis" should be substituted with "Nanoparticle Tracking Analysis".
4) Line 96: authors should specify which EV profile they dissected with the MACSPlex kit.
5) Authors tested FBS deprivation and different doses of tunicamycin to induce ER stress. Regarding tunicamycin, they detected a dose-dependent effect on CHOP and GRP78 transcript levels (fig. 3). However, in the next experiments, they first stuck to the lowest concentration (experiments showed in fig. 4) and then abandoned tunicamycin in favor of cell starvation (experiments showed in fig. 5, 6, and 7). Is there a specific reason? If so, could authors better specify it in the manuscript? 
6) caption of figure 5a-b. The term “micrograph” is normally used to refer to microscopy images, not western blot membranes. Consider correcting it.
7) figure 6. I would expect graphs in fig. 6a and 6b to be almost specular. I believe there was an error when plotting the graphs or axis tags or in data analysis/normalization. How is possible that a negative control has 100% of apoptotic cells (fig. 6a)? Did I misinterpreted the data?
8) Caption of figure 7c. The authors claimed to have used RNU6b as a control, and GAPDH expression to normalize miRNA data. Would be possible to include RNU6b normalized level in the graphs?
9) Materials and methods: Overall English language of this section could be improved. I find that it is a bit poor compared to the introduction, results, and discussion sections.
10) line 292: was the FBS used EV-free? Could the authors specify it?
11) the catalog # of the antibodies is missing in section 4.8.
12) line 396: the sentence is truncated.
13) line 409-410: what “proper endogenous controls” were used for the normalization in miRNA analysis? Was GAPDH expression as stated in the caption of fig. 7c?
14) line 325: is the incubation temperature correct (-4°C)?
15) Caption of figure S1. English language should be improved.

Major concerns

My first concern regards statistical analysis.
For RT-PCR experiments, authors claimed they normalized the expression of each transcript on GAPDH expression, and then they further normalized treated samples results on control sample one. So, the values plotted on the graphs in figs. 3, 4, and 7c should refer to the ratios of transcript X expression/GAPDH expression normalized on control (am I correct?).

  • Did the authors perform statistical analysis before or after further normalizing the data on CTL? Could authors specify it?
  • Did the authors check if GAPDH expression was consistent and similar between the samples used in each experiment? If so, could you add a figure/mention it?
  • Authors claim that the results come from at least 3-4 independent experiments, and data are plotted as mean ± SD. However, the CTL sample has no error bar. Does it mean that SD on CTL is absent/negligible? Or that CTL was not analyzed in triplicate/quadruplicate? Or, again, that CTL SD was not considered, and the normalized value ”1” was assigned to it regardless of SD? The difference is subtle but could lead to major differences in statistical results.

I’m asking all this because some of the statistical analysis authors performed is far from clear to me. I am not a super expert in statistics, but I noticed some unexpected things. For example, in fig.4:

1) The test applied here was Student’s t-test, isn’t it? The caption does not specify it.
2) fig. 4a and 4b. In fig. 4a, grey vs. black gives p < 0.0001. The same analysis performed in fig. 4b gives p < 0.01. However, this is strange since the difference between grey and black in fig. 4b is HIGHER (and SD is smaller) than the one between grey and black in fig. 4a. Thus, the p should be smaller in fig. 4b than in fig. 4a, not the opposite. This point of course applies if the statistical analysis was performed after the normalization of samples on CTL.
3) fig. 4c. The difference between cyan and grey appears to be significant. However, I find it strange that the one between red and grey does not, since red is lower than cyan (and even SD is smaller). The same goes for fig. 4e: cyan vs. grey is significant, red vs. grey is not. Again, this point of course applies if the statistical analysis was performed after the normalization of samples on CTL.
4) fig. 4h. Like the previous one. Brown vs grey is significant, but blue vs grey is not, although blue is averagely smaller than brown. Again, this point of course applies if the statistical analysis was performed after the normalization of samples on CTL.

My second concern regards figure 7.

Fig. 7a depicts a diagram of some of the miRNAs involved in the regulation of desired ER stress-related genes. Some of them can be also found in MSCs EVs (the ones in the black box). However, I cannot understand what the numbers stand for. Do they refer to miRNA codes (e.g. 32 = miRNA-32, 193 =miRNA-193, etc.)? Or do they refer to the “number of miRNAs that were found to target the indicated transcript”?  Because after reading the caption I would propend for the first option. If so, I cannot get the meaning of fig 7a, and its link with fig.7b. Indeed, in fig. 7b, a list of miRNAs able to target the transcripts of interest are shown, together with their relative abundance in MSC- and serum-derived EVs. However, none of the miRNAs in fig.7a have been analyzed.

Otherwise, if the second option was intended in fig.7a (“number of miRNAs found to target the indicated transcript”), the caption of fig. 7a is misleading and should be modified.

Spell check

1) Line 74: "of" should be removed.
2) Caption of Fig. 3. Tunicamycin range of concentration is reported to be 0.4 - 1.6 ug/mL. However, in the caption of fig. 4, the concentration becomes 0.4 ng/mL (1000 times smaller than the smallest concentration tested). Is it intentional? If not, please correct the wrong one.
3) Line 277: I think “road” was intended instead of “rode”.
4) caption of figure 7c. “by real-time PCR” is not necessary, as it was specified at the beginning of the sentence.

Author Response

Please see the uploaded document

Reviewer 2 Report

This interesting manuscript investigates the effects of MSC-EVs on the processes of apoptosis at the in vitro models of corneal dystrophy. The authors performed a bunch of analyses to characterise EVs, describe the model used and investigate the effects of MSC-EVs on the mRNA levels and protein levels of apoptosis factors and, finally, analyzed potential rescue of cells from apoptosis after treatment with EVs. MSC-EVs were compared with serum EVs from ischemic donors, it is a good control for this study. Although the manuscript seems to be well-performed and technically sound, the authors overstate the significance of the observed effect of EVs on the rescue of cells from apoptosis. The numbers from their data say that the EVs slightly ameliorate apoptosis and weakly affect apoptosis rates. I strongly suggest the authors to mitigate their statements in the abstract and throughout the text. The rest of the comments are listed below:

The graphical abstract gives an impression that MSC-EVs induce apoptosis

Figure 1B - control missing for CD166 staining

Line 92 - the authors should mention the method they used for isolating EVs so that the readers do not have to find and read the references

Nanosight analysis can be moved from Supplementary to the body of the manuscript

Figure 4: the authors should mention that they are comparing the tun/2% group with other groups, or get rid of the control group here as it does not provide any information

Figure 6: in fact, the effect of msc-evs on apoptosis is very weak in contrast to what is written in the abstract and the body of the manuscript. The decrease from 100 to around 80% with big SD means that the cells are not rescued. It is totally instrumental for the correctness of the paper to state it clearly and not over-interpret and overestimate the data

What is the amount of exosomes that was used for the study?

The experimental design is not clear and not presented. As such, it is hard to understand when the effects occur, how the treatments are added etc

it is also not clear from the paper, why the authors performed such an exhaustive description of extracellular vesicles. These analyses provide in-depth characterisation, which is commonly not necessary 

Author Response

Please see the uploaded responce letter

Round 2

Reviewer 2 Report

The authors addressed my concerns and substantially revised the manuscript, its quality improved and it can now be considered to publication.